# New TMA (4,6,4′-Trimethyl angelicin) Analogues as Anti-Inflammatory Agents in the Treatment of Cystic Fibrosis Lung Disease

**DOI:** 10.3390/ijms232214483

**Published:** 2022-11-21

**Authors:** Chiara Tupini, Adriana Chilin, Alice Rossi, Ida De Fino, Alessandra Bragonzi, Elisabetta D’Aversa, Lucia Carmela Cosenza, Christian Vaccarin, Gianni Sacchetti, Monica Borgatti, Anna Tamanini, Maria Cristina Dechecchi, Francesca Sanvito, Roberto Gambari, Giulio Cabrini, Ilaria Lampronti

**Affiliations:** 1Department of Life Sciences and Biotechnology, University of Ferrara, Via Fossato di Mortara 74, 44121 Ferrara, Italy; 2Department of Pharmaceutical and Pharmacological Sciences, University of Padova, Via Marzolo 5, 35131 Padova, Italy; 3Center of Innovative Therapies for Cystic Fibrosis (InnThera4CF), University of Ferrara, 44121 Ferrara, Italy; 4Infections and Cystic Fibrosis Unit, Division of Immunology, Transplantation and Infectious Diseases, IRCCS San Raffaele Scientific Institute, Via Olgettina 60, 20132 Milano, Italy; 5Center for Radiopharmaceutical Sciences ETH-PSI-USZ, Paul Scherzer Institute, 5232 Villigen, Switzerland; 6Department of Neurosciences, Biomedicine and Movement, Section of Clinical Biochemistry, University of Verona, Piazzale Stefani 1, 37126 Verona, Italy; 7Pathology Unit, Division of Experimental Oncology, Histopathology Laboratory of GLP Test Facility, San Raffaele Telethon Institute for Gene Therapy (SR-Tiget), IRCCS San Raffaele Scientific Institute, 20132 Milan, Italy

**Keywords:** cystic fibrosis, inflammation, CFTR (Cystic Fibrosis Transmembrane conductance Regulator), anti-inflammatory agents, IL-8 (Interleukin 8), TMA (trimethyl angelicin) derivatives, pre-clinical studies

## Abstract

A series of new-generation TMA (4,6,4′-trimethyl angelicin) analogues was projected and synthetized in order to ameliorate anti-inflammatory activity, with reduced or absent toxicity. Since the NF-κB transcription factor (TF) plays a critical role in the expression of IL-8 (Interluekin 8), a typical marker of lung inflammation in Cystic Fibrosis (CF), the use of agents able to interfere with the NF-κB pathway represents an interesting therapeutic strategy. Through preliminary EMSA experiments, we identified several new TMA derivatives able to inhibit the NF-κB/DNA complex. The selected active molecules were then analyzed to evaluate the anti-inflammatory effect using both *Pseudomonas aeruginosa* (PAO1) infection and TNF-alpha stimulus on the CF IB3-1 cell line. It was demonstrated that mainly two TMA analogues, GY971a mesylate salt (6-*p*-minophenyl-4,4′-dimethyl-angelicin) and GY964 (4-phenyl-6,4′-dimethyl-angelicin), were able to decrease the IL-8 gene expression. At the same time, these molecules were found to have no pro-apoptotic, mutagenic and phototoxic effects, facilitating our decision to test the efficacy in vivo by using a mouse model of acute *P. aeruginosa* lung infection. The anti-inflammatory effect of GY971a was confirmed in vivo; this derivative was able to deeply decrease the total number of inflammatory cells, the neutrophil count and the cytokine/chemokine profile in the *P. aeruginosa* acute infection model, without evident toxicity. Considering all the obtained and reported in vitro and in vivo pre-clinical results, GY971a seems to have interesting anti-inflammatory effects, modulating the NF-κB pathway, as well as the starting lead compound TMA, but without side effects.

## 1. Introduction

Cystic fibrosis (CF) is a progressive, lethal, autosomal recessive monogenic disease, caused by defects in a single gene, Cystic Fibrosis Transmembrane conductance Regulator (CFTR), which encodes for a chloride and bicarbonate channel expressed in several epithelia. The inheritance of mutations of the CFTR gene is at the basis of the multiorgan CF disease [1,2,3].

Pulmonary manifestations are responsible for most of the morbidity and mortality. The absence of CFTR-mediated chloride and bicarbonate secretion has been functionally linked to airway surface dehydration, which leads to the accumulation of thick mucus, increased susceptibility to infection and inflammation. CF lung disease, characterized by chronic bacterial airway infection, neutrophilic inflammation and dilation of bronchioles obstructed by mucus plugs, is presently the main limitation to the quality and expectancy of life of CF patients. A consensus has been reached that lung pathology initiates in the early months of life, often before the onset of clinical symptoms, as shown by the presence of inflammatory cytokines in the bronchoalveolar lavage fluid (BALF) [4,5,6] and by the lung histopathology of CF infants who died within weeks or months of birth, showing bronchial lumen filled and plugged by neutrophils [7]. The CFTR protein reduces the hydration and the pH of the airway surface liquid (ASL), thus affecting the rate of mucociliary clearance, the principal innate mechanism involved in the defense against microbial infection [8].

Hallmarks of the lung pathology of CF patients are defective mucociliary clearance, as well as chronic bacterial infection (especially *Pseudomonas aeruginosa*), associated with an exaggerated neutrophil-dominated inflammation. Although new potent small-molecule modulators rescuing the mutant CFTR protein, such as the CFTR modulator Trikafta, provide substantial benefit to patients carrying the most common F508del-CFTR mutation, lung inflammation persists, resulting in progressive tissue damage. Different strategies have been proposed to tackle CF lung inflammation [9]. One of these approaches is based on the regulation of the recruitment on neutrophils inside CF airway lumen, targeting different molecules, including the main neutrophil chemokine Interleukin 8 (CXCL8/IL-8) [10].

Neutrophils are the predominant immune cells infiltrating the airway mucosa and filling the intraluminal space of the bronchioles of CF patients [7]. Neutrophils are unable to solve CF bacterial infection. Although defective in clearing the chronic respiratory infection of these patients, neutrophils in CF airways are exposed to bacteria and become a source of continuous release of proteases [11]. A second critical adverse effect of the huge number of neutrophils is their contribution to increasing the pro-oxidant milieu of the CF ASL, as extensively reviewed elsewhere [12]. As a third critical adverse effect, the presence of many neutrophils in CF bronchial lumen implies the release of abundant DNA on the surface of the mucosa, which further reduces the fluidity of the ASL and worsens the bronchial obstruction [13]. In synthesis, the regulation of the excessive recruitment of neutrophils in CF airway lumen mediated by CXCL8/IL-8 to a level that reduces tissue damage without abrogating innate immune defenses is a key anti-inflammatory approach in CF [14].

Although drugs that target the CFTR have recently been approved and show great promise, it is still not clear how CFTR modulators affect the infection and inflammation, and the research on novel anti-inflammatory agents has been obscured by the discovery of CFTR modulators, thereby inflammation represents “a sort of orphan target in CF” [15]. In fact, no anti-inflammatory agents specific for CF lung disease have been developed, and those commercially available, steroidal and non-steroidal anti-inflammatory drugs, are characterized by known and important side effects. Thus, it is interesting to improve studies and protocols for targeting the host response. Since the NF-κB transcription factor (TF) plays a critical role in IL-8 expression, the use of agents designed to interfere with the NF-κB pathway represents an interesting therapeutic strategy [10].

All our research was aimed at designing and developing molecules capable of acting as anti-inflammatories, inhibiting the NF-κB pathway, and possibly also as CFTR correctors, in order to obtain in the future a single drug with double activity.

Since the research of modern therapies to combat inflammation in CF is still aimed at finding new potential anti-inflammatory drugs with different mechanisms of action that may supplement or replace the use of current drugs to limit known side effects, we expanded our research to new synthetic analogues, structurally related to TMA (4,6,4′-trimethyl angelicin) [16,17], in order to analyze and develop new drugs, hoping they may provide real therapeutic options for CF inflammation.

The aim was to modify the original TMA scaffold (Figure 1) in order to identify new compounds with biological properties comparable to TMA but with reduced or absent side effects.

Therefore, a small library of derivatives was recently synthesized and studied [18,19]. Different substituents were inserted at positions 4, 6, or 4′ of the tricyclic nucleus to evaluate the role of these positions in the overall activity of the compounds. In detail, bulky substituents, such as hindered alkyls or phenyl, at position 4, abolished photoreactivity towards DNA, as well as the introduction of hindered alkyls or aryl at position 6. All these modifications also allowed for the prevention of unwanted photoreactivity and mutagenicity of the parent TMA. Regarding the CFTR modulation activity, the insertion of hindered substituents at position 4, such as the isopropyl group, preserved the CFTR correction and potentiation properties, while the introduction of bulky substituents at position 6 was detrimental for potentiation. Few or no modifications in the furan ring were tolerated, as the CFTR correction properties were maintained only in the presence of a methyl group at position 4′.

In the present work, we selected some of these new TMA analogues to investigate their anti-inflammatory activity (Figure 2).

## 2. Results and Discussion

The selected TMA derivatives were first analyzed in vitro, with EMSA experiments and in human IB3-1 bronchial epithelial cells, and then the best analogues were utilized for pre-clinical studies on a murine model of *P. aeruginosa* respiratory infection.

### 2.1. Synthesis of TMA Analogues

The TMA analogues were synthesized as previously reported [18,19]. The GY971 derivative (6-p-aminophenyl-4,4′-dimethyl-angelicin, also called pANDMA), is characterized by an aniline moiety at position 6 of the furocoumarin structure (Figure 2). The amino group, salified as mesylate salt, allowed the formation of a water-soluble derivative: the hydrophilic mesylate, named GY971a, demonstrated to be four-times more bioavailable than the parent GY971 [19]. Instead, typically all the TMA analogues were lipophilic and solubilize only in DMSO. This peculiarity of the GY971a derivative makes the molecule interesting for the possible development of a drug that can be administered via aerosol. The effect of GY971 was also confirmed using the relative mesylate GY971a and, for all the following experiments in vitro and in vivo, the salified form was always used.

### 2.2. In Vitro Experiments

#### 2.2.1. NF-κB Inhibition Activity of TMA Analogues (EMSA)

Preliminary electrophoretic mobility shift assay (EMSA) experiments, depicted in Figure 3, were performed on different TMA analogues. GY955, GY956, GY964, GY966 and GY971 (pANDMA) were found able to inhibit the NF-κB/DNA complex. The obtained data, summarized in Table 1, demonstrate that, when used at 12.5 μM concentration, the GY971 derivative fully suppresses the interactions between NF-κB p50 and the specific target DNA, demonstrating the best activity, while other molecules (compounds GY957 and GY967) showed low activity (MIC-Inhibition of NF-κB/DNA complex- >100 μM).

#### 2.2.2. NF-κB-Mediated IL-8 Expression and Release after Treatment with TMA Analogues

The newly synthetized analogues were characterized (before the activation of the in vivo assays) in order to investigate the IL-8 (Interleukin 8) gene expression in human IB3-1 bronchial epithelial cells, infected with *P. aeruginosa* (PAO1 strain) or stimulated by the Tumor Necrosis Factor-alpha (TNF-α). The activity of the synthesized molecules was studied by RT-qPCR and ELISA analysis.

##### Effect of TMA Analogues on IL-8 Expression in IB3-1 F508del/W1282X Cells Infected with *P. aeruginosa*

The in vitro analyses were conducted on IB3-1 cells, heterozygous for the F508del and W1282X mutations. When cells were infected with the *P. aeruginosa* PAO1 strain, mainly GY964 (4-PhDMA) and GY971a (pANDMA) at 200 nM were demonstrated to significantly inhibit IL-8 expression (biomarker of inflammation in CF), showing a 86.5% and 52.4% inhibition, respectively (Figure 4A); this activity was also confirmed at lower concentrations (50–100 nM), as reported in Figure 4B (30.7% and 50.4%, respectively) and Figure 4C (20.5% and 45.5%, respectively).

##### Effect of TMA Analogues on IL-8 Expression in TNF-α-Stimulated IB3-1 508Fdel/W1282X Cells

The best TMA analogous, GY964 (4-PhDMA) and GY971a (pANDMA), were also analyzed on TNF-α-stimulated IB3-1 cells, another CF inflammation in vitro model useful for the study of potential anti-inflammatory drugs. Both GY964 (Figure 5A) and GY971a (Figure 5B) inhibited the transcript for IL-8, even at low concentrations (50 and 100 nM), as shown in Figure 5A (32.4% and 64.9%, respectively) and in Figure 5B (21.1% and 36.2%, respectively).

##### IL-8 Release in IB3-1 Cells Stimulated with TNF-α or Infected with *P. aeruginosa*

IL-8, secreted into the medium during the TNF-α-stimulated or PAO1-infected IB3-1 cell culture, was quantified in the presence of TMA analogues. IL-8 release was in agreement with that obtained studying IL-8 mRNA accumulation. This is shown in Figure 6, which supports the decrease in IL-8 synthesis after the treatment with GY964 (4-PhDMA) and GY971a (pANDMA). We may observe a reduction of between 36 and 38% in IL-8 secretion when TNF-α-stimulated IB3-1 cells were treated with GY971a (100 and 200 nM) compared to untreated (Figure 6B), and of between 77 and 87% when PAO1-infected IB3-1 cells were treated with GY971a at the same dosages (Figure 6D). GY964 (4-PhDMA) was also demonstrated to be a good inhibitor of IL-8 synthesis at lower concentrations (50, 100 and 200 nM) only in the PAO1-infected IB3-1 cells (Figure 6C), showing an IL-8 inhibition of between 58 and 75%.

#### 2.2.3. Toxicity Studies (Antiproliferative and Apoptotic Activity Evaluation)

The known major toxic effects of TMA are related to its ability to bind, under UVA irradiation, DNA strands, with the detection of photoadducts. As recently shown by our research group (Vaccarin et al.) [19], only TMA, used as a reference, showed photoadduct formation, while the other compounds, GY964 (4-PhDMA) and GY971 (pANDMA), selected on the basis of their anti-inflammatory activity (Figure 4), presented only the unchanged bands detected with TLC analysis, thus demonstrating that the two newly synthesized compounds do not react with DNA under UVA irradiation (Appendix A), and no genotoxicity was observed either [19].

In order to exclude any unexpected activities, different assays were performed to evaluate the possible toxic effects of the selected TMA analogues, also analyzing the cell growth, apoptosis (Figure 7A,B) and anti-bacterial activity (Figure 7C).

The cell growth and apoptosis analysis, conducted on IB3-1 and on CFBE41o- cell lines (Figure 7A,B) and performed using GY964 and GY971a derivatives, was managed at three different concentrations (50, 100 and 200 nM), comparing treated cultures with untreated cells (C-) and parent TMA. Representative data, obtained from the analysis of the corresponding Muse Analyzer plots 48 h after the treatment, are shown in Figure 7A (on IB3-1 cell line) and Figure 7B (on CFBE41o- cell line) and are shown in detail in Table 2 and Table 3, which show the complete set of results obtained studying the effects of these compounds in early and late apoptosis stages. No cell death or pro-apototic effects were observed (Figure 7A,B), even at the highest concentrations (200 nM). The Muse Analyzer dot-plots of all treatments are shown in additional figures (Appendix A).

To exclude the possibility that the inhibition of *P. aeruginosa*-dependent induction of IL-8 is due to an indirect anti-bacterial effect on *P. aeruginosa*, we evaluated the effects of GY964 (4-PhDMA) and GY971a (pANDMA) on *P. aeruginosa* (PAO1 strain) cultures, as shown in Figure 7C. The GY955, GY956, GY964, GY966 and GY971 compounds were also assayed for genotoxicity evaluation through the Ames test [19], and no genotoxic activity emerged for any of the samples at all the concentrations tested, with all the Salmonella strains.

### 2.3. In Vivo Experiments

#### In Vivo Toxicity and Efficacy of TMA Analogue GY971a (pANDMA) in a Murine Model of *P. aeruginosa* Acute Infection

The most active selected TMA analogues, GY964 (4-PhDMA) and GY971a (pANDMA), were tested in a murine model of *P. aeruginosa* acute lung infection. Before the GY971a efficacy studies, C57Bl/6NCr male mice were treated with three different doses of TMA analogue (4.5, 9 and 18 mg/kg) or a vehicle aerosol by Penn Century to evaluate the toxicity. The delivery method was chosen to be representative of aerosol delivery in humans, and the method was selected to ensure that the entire dose directly nebulized into the murine lung. Following 24 h after treatment, several readouts, including weight loss, body temperature and general health status, did not show significant changes between GY971a and the vehicle. A histopathological evaluation of liver, spleen, kidneys, bone marrow, stomach, trachea and lungs was performed. No pathological changes related to acute toxicity were observed following the administration of GY971a. Microscopic changes were recorded in the lungs, without a difference in the incidence and severity of lesions, and were considered related to damage due to the intratracheal inoculum of PBS. The changes in the bone marrow, spleen and liver were sporadic findings considered incidental and observed throughout all groups.

Next, C57BL/6NCrlBR mice were challenged with a 1 × 10^6^ planktonic *P. aeruginosa* PAO1 strain by intratracheal inoculation to induce acute infection [20,21]. Local treatment via the aerosol route with 4.5, 9 and 18 mg/kg GY971a started 5 min after infection and was compared with the vehicle (H_2_O/DMSO 4%).

After 6 h from acute infection, mice were monitored for abnormal behavior, weight loss, gross visual evaluation of health status and body temperature. The bacterial loads and inflammatory response in the lung homogenate and bronco-alveolar lavage fluid (BALF) were evaluated (Figure 8).

The best results were obtained using the treatment with GY971a (pANDMA), while the treatment with GY964 (4-PhDMA) did not produce interesting effects; in fact, the derivative did not induce a reduction in the total number of cells and neutrophils, and a not significant reduction in CFUs (Colony Forming Units) was observed.

To define the effect of GY971a on the airway inflammatory response, we measured leukocyte recruitment in the BALF 6 h after infection. A single dose of TMA analogue reduced the total number of cells, particularly neutrophils, in a dose-dependent manner, with statistical significance at all doses compared to the vehicle (Figure 9A,B), indicating a reduction in inflammation. In addition, a significant reduction was observed in macrophage numbers after GY971a treatment at a higher dose (Figure 9C).

To verify that reducing inflammatory cells in infected mice does not impair host defense or exacerbate infection, the bacterial load was evaluated in the airways of mice, including the BALF and lung. No significant increase in bacterial burden was detected in the CFUs between GY971a and the vehicle (Figure 9C–E).

To confirm the effect of GY971a on the airway inflammatory response, the cytokine/chemokine concentrations in the lung homogenates were evaluated.

Mice treated with 9 and 18 mg/kg GY971a showed a significant reduction in the typical chemokines activated by bacterial infection, such as MCP-1 and MIP-1α, in the lung when compared to vehicle-treated mice. IL-1α and IL-1β, the pro-inflammatory cytokines that play a central role during infection were also significantly decreased in the treated mice compared to the vehicle-treated mice (Table 4). Other cytokines/chemokines were reduced after aerosol treatment with GY971a, including IL-10, IL-12p40, IL-12p70, IL-17A, INF-γ, granulocyte–macrophage colony-stimulating factor (GM-CSF) and granulocyte colony-stimulating factor (G-CSF) (Table 4).

Our results indicate that a single dose of GY971a treatment by aerosol was effective in reducing acute airway murine inflammation induced by *P. aeruginosa* in a dose-dependent manner.

## 3. Materials and Methods

### 3.1. Synthesis of TMA Derivatives

The TMA analogues were prepared as previously described [18,19]. GY964, GY955, GY956, GY966 and GY971 were dissolved in 100% DMSO. GY971a (mesylate pANDMA) was initially dissolved in H_2_O + 4% DMSO to obtain 20 mM stock solution; all the subsequent dilutions (working solutions) used for the experiments were obtained in H_2_O. Tobramycin (Merck KGaA, Darmstadt, Germany) was dissolved in 100% H_2_O.

### 3.2. In Vitro Assays

#### 3.2.1. EMSA Experiments

EMSA (Electrophoretic Mobility Shift Assay) was performed as previously described [16]. Briefly, double-stranded synthetic oligodeoxynucleotides mimicking the NF-κB binding (NF-κB, sense: 5′-CGC TGG GGA CTT TCC ACG G-3′) were employed. Oligodeoxynucleotides were labeled with [γ-32P]ATP using 10 units of T4-polynucleotide-kinase (Fermentas, Thermo Fisher Scientific, Waltham, MA, USA) in 500 mM Tris·HCl, pH 7.6, 100 mM MgCl_2_, 50 mM dithiothreitol, 1 mM spermidine, 1 mM EDTA in the presence of 50 mCi [γ-32P]ATP in a volume of 20 μL for 45 min at 37 °C. The reaction was brought to 150 mM NaCl, and 150 ng complementary oligodeoxynucleotide were added. Reaction temperature was increased to 100 °C for 5 min and left decreasing to room temperature overnight. Binding reactions were set up in a total volume of 20 μL containing buffer plus 5% glycerol, 1 mM dithiothreitol, 10 ng of human NF-κB p50 protein (Promega, Madison, WI, USA) and different concentrations of compounds. After an incubation of 20 min at room temperature, 0.25 ng of 32P-labeled oligonucleotides was added to the samples for a further 20 min at room temperature and then they were electrophoresed at constant voltage (200 V) under low ionic strength conditions (0.25 × TBE buffer: 22 mM Tris-borate, 0.4 mM EDTA) on 6% polyacrylamide gels. Gels were dried and subjected to standard autoradiographic procedures.

#### 3.2.2. Gene Expression Analysis Induced by TMA Analogues on PAO1-Infected or TNF-alpha-Stimulated IB3-1 Cells

Cell culture conditions, TMA analogue treatment, infection with *Pseudomonas aeruginosa* or addition of TNF-alpha stimulus:

IB3-1 cells, derived from a CF patient with a ΔF508/W1282X mutant genotype and immortalized with adeno12/SV40, were cultured in LHC-8 supplemented with 5% FBS in the absence of gentamycin in a 5% CO_2_ humidified atmosphere incubator at 37 °C. Cell passage was performed two times a week. For the experiments, confluent cells were rinsed with PBS, detached using Trypsin-EDTA (4 min at 37 °C (5% CO_2_), neutralized with FBS and resuspended in LHC-8 medium. At this step, 50 µL of cells diluted in 5 mL of physiological solution were counted using a coulter counter (BECKMAN COULTER^®^ Z Series). Then, 50 × 10^3^ cells/well were seeded in a 12-well plate (1 mL/well) and incubated a 37 °C (5% CO_2_) [22].

To perform our experiments, the PA0-1 strain was utilized, which is the most common used for research. The prototype used was produced by A. Prince (Columbia University, New York, NY, USA) and it is a non-mucoid strain suited for the laboratorial technique because of its mild effects on operators. As indicated in the infection protocol described below, we used both the proper growth mediums: trypticase soy broth (TSB) or agar (TSA) (Difco). Although *P. aeruginosa* is an aerobic–anaerobic facultative bacterium, the growth is facilitated in anaerobic conditions, so we sealed the tubes during the time of replication. Our aliquots of PA0-1 were stocked in glycerol at −80 °C [22].

GY964, GY955, GY956, GY966, GY971 and GY971a were dissolved in 100% DMSO. GY971a (pANDMA) was initially dissolved in H_2_O + 4% DMSO; all the subsequent dilutions used for the experiments were obtained in H_2_O.

Regarding the treatment with TMA derivatives and stimulation with TNF-α, 50.000–100.000 cells/mL of IB3-1 cells were seeded in 12-well plates in LHC-8 medium in the presence of 5% FBS. After 48 h from seeding, the compounds were added, 5 h before stimulation with TNF-α 100 ng/mL, and incubated for a further 24 h. After that, the supernatants were collected, and the total RNA was extracted [23].

Regarding the treatment with TMA derivatives and *P. aeruginosa* (PAO1) infection, 50.000–100.000 cells/mL of IB3-1 cells were seeded in 12-well plates in LHC-8 medium in the presence of 5% FBS. After 48 h from seeding, the compounds were added, the day before infecting with PA. Aliquot of overnight *P. aeruginosa* culture was suspended in TSB, adjusting the volume and the *P. aeruginosa* amount to reach 0.5 McFarland Units (MCFU) as its initial density, which corresponds to 1.5 × 10^−8^ cells/mL. The bacterial suspension was incubated at 37 °C with shaking until the start of the exponential-growth phase, which corresponds to double the initial bacterial density. The bacterial suspension was centrifuged twice at 4 °C, 7000 rpm for 10 min (Allegra 64R, Beckman Coulter, Brea, CA, USA) and, finally, diluted 1:10 in DPBS. The supernatants above the pre-treated cells grown in plates were removed, to add the volumes of fresh culture mediums (LHC-8 without FBS) and of bacterial suspension. In order to study the acute inflammation, we incubated the plates for 4 h. After that, the supernatants were collected, and the total RNA was extracted.

Total RNA was extracted using TRIzol Reagent (Sigma-Aldrich, St. Louis, MO, USA) following the manufacturer’s instructions. The RNA spectrophotometric quantification was performed by using a single UV ray spectrophotometer SmartSpec™ Plus (BioRad; Hercules, CA, USA). Each sample was then prepared diluting 1 µL of RNA with 50 µL of R.F. (RNAse-Free) water in a cuvette and then analyzed. The concentration of the extracted RNA was obtained calculating the proportion: 1 OD260: 40 ng/µL. Reverse transcription (RT) was performed using a Reverse Transcription System kit (Promega, Madison, WI, USA): 500–1000 ng of total RNA was reverse transcribed adding 13.5 μL of a mix containing: 5 mM MgCl_2_, Buffer 1× (RT Buffer 10×, Promega), dNTPs Mix 50 µg/mL (Takara-Bio, San Jose, CA, USA), recombinant RNasin^®^ 40 units/µL (Promega, USA), ImProm-II™ Reverse Transcriptase, and 1μL Random Primers (Promega, USA), diluted when necessary with H_2_O R.F. in a total volume of 20 μL. The reaction was performed using the following protocol at the Thermal Cycler (GeneAmp^®^ PCR System 9700, Applied Biosystem, Waltham, MA, USA): 5 min at 25 °C, 1 h at 42 °C, 15 min at 70 °C. The cDNAs were finally stocked at −20 °C. The oligonucleotides used as primers to amplify the IL-8 and Glyceraldehyde 3-phosphate dehydrogenase (GAPDH) genes were produced by IDT (Integrated DNA Technologies, Coralville, IA, USA) and by Sigma-Aldrich. Primer sequences were drawn up through the Primer-BLAST software: GAPDH forward 5′-AAG GTC GGA GTC AAC GGA TTT-3′, GAPDH reverse 5′-ACT GTG GTC ATG AGT CCT TCC-3′, IL-8 forward 5′-GTG CAG TTT TGC CAA GGA GT-3′, IL-8 reverse 5′-TTA TGA ATT CTC AGC CCT CTT CAA AAA CT-3′.

Gene expression analysis:

The resulting cDNA was quantified by relative quantitative real-time PCR (real-time qPCR). For real-time qPCR, 1 μL of cDNA was used for each SYBR Green reaction to quantify the relative expression of IL-8. Real-time PCRs were performed using an iTaq Universal SYBR Green Supermix (Bio-Rad Laboratories Inc., Hercules, CA, USA). Real-time PCRs were performed for 40 cycles with the following protocol: denaturation, 95 °C for 5 s; annealing, 55 °C–95 °C for 5 s; elongation, 60 °C for 30 s. The relative proportions of each amplified template were determined utilizing the threshold cycle (Ct) value for each performed PCR. The ddCt method was used to compare gene expression data. Each sample was quantified in duplicate. Changes in mRNA expression level were calculated following normalization with the GAPDH calibrator gene (housekeeping gene) and expressed as fold change over untreated samples [24].

Statistical analysis

All the results were expressed as the means ± SD. Statistical analysis was performed by parametric (one-way ANOVA) and non-parametric (Mann–Whitney) tests using the GraphPad Prism 8.2.1 software (GraphPad Software, Inc., La Jolla, CA, USA). A *p* value of <0.05 was considered statistically significant.

#### 3.2.3. Protein Expression Analysis Induced by TMA Analogues (ELISA)

The synthesis of IL-8 in IB3-1 cells induced by PA0-1 and TNF-α and modulated by TMA analogues was evaluated through the Human CXCL8 Pre-Coated Enzyme-linked immunosorbent (ELISA) assay kit (PicoKine™; Boster-Bio, Pleasanton, CA, USA). To perform the experiment, we recovered the supernatants stocked at −80 °C of infected/stimulated IB3-1 cells with PA0-1/TNF-α and then treated with compounds. Standards, obtained by serial dilutions, the blank and samples were added to the wells; then, we incubated with the biotinylated detection antibody. After washing the wells with TBS buffer, the Avidin–Biotin–Peroxidase Complex (ABC-HRP) was incubated. The plate was washed five times, and the color-developing reagent (TMB) was added. Then, the 96-well plate was analyzed at the Sunrise Microplate Reader (Tecan, Hombrechtikon, Switzerland) and data were acquired through the Magellan™ software (Tecan, Hombrechtikon, Switzerland). Finally, the concentration of human CXCL8 in each sample was calculated using the standard curve obtained by plotting the absorbance intensity (45 m) with the standard concentrations.

#### 3.2.4. Toxicity

Cell Proliferation Assay

IB3-1 cells were seeded in 24-well plates in LHC-8 medium in the presence of 5% FBS. CFBE41o- cells were seeded in 24-well plates in MEM medium supplemented with 10% FBS, 2 mM L-glutamine and 2 µL/mL puromycin. Human bronchial epithelial cells, CFBE41o- cells, stably transducted with ΔF508-CFTR, were maintained in MEM supplemented with 10% FBS, 2 mM L-glutamine and 2 µL/mL puromycin in a 5% CO_2_ 95% air incubator at 37 °C. At day one, the cells were seeded in cell-culture 24-well plates at a density of 45 × 10^3^ cells/well in MEM supplemented with 10% FBS and 2 mM L-glutamine for the experiments.

After 48 h from seeding, the compounds were added. After 24 and 48 h from treatment, cells were rinsed with PBS, detached using Trypsin-EDTA (4 min at 37 °C, 5% CO_2_), neutralized with FBS and resuspended in LHC-8 medium. At this step, 50 µL of cells diluted in 5 mL of physiological solution were counted using a coulter counter (BECKMAN COULTER^®^ Z Series). The IC_50_ value (the concentration of compound leading to 50% inhibition of cell growth) was calculated after 24 h and 48 h of culture [25].

Proapoptotic activity (Annexin V method)

IB3-1 and CFBE41o- cells were treated with different concentrations of GY971a (pANDMA) and GY964 (4-PhDMA) in order to evaluate possible apoptotic effects. To obtain 200, 100 and 50 nM concentrations, 2.5 μL of 80, 40 and 20 μM working solutions were added, respectively, to treat 1 mL of cell cultures (in 24-well plates).

Annexin V and Dead Cell assays on untreated and treated cells were performed with the Annexin V and Dead Cell reagent and the Muse cell analyzer (Millipore-Luminex, Billerica, MA, USA), according to the instructions supplied by the manufacturer. This procedure utilizes Annexin V to detect PS (Phosphatidyl Serine) on the external membrane of apoptotic cells. Four populations of cells can be distinguished using this assay: live, early apoptotic, late apoptotic and dead cells. The MUSE cell analyzer is able to discriminate “early” from “late” apoptosis since “early apoptotic cells” are Annexin V-positive, and 7-AAD-negative, while “late apoptotic cells” are positive to both Annexin V and 7-AAD (7-aminoactinomycin D). Cells were washed with sterile PBS 1×, tripsinized, resuspended in the original medium and diluted (1:2), with the one-step addition of the Muse Annexin V and Dead Cell reagent. After incubation for 20 min at room temperature, samples were analyzed, using Triton X 0.01%, as a positive control [25].

Antimicrobial activity (Broth Dilution Method)

One single PAO1 colony was diluted in 7 mL TBS medium. 1.4 µL of PAO1 suspension was added in 2 mL TBS/tube. One tube contained only 2 mL of TBS as a negative control. The compounds were added at different concentrations (50, 100, 200 nM). The concentration of PAO1 was measured using the densitometer (DEN-B, Grant Bio, Keison products, UK) every 2 h for a total period of 6 h. The ratio of the MCFU values obtained after 6 h/the MCFU values at time 0 h were obtained and plotted [26].

Statistical analysis

All the results were expressed as the means ± SD. Statistical analysis was performed by parametric (one-way ANOVA) and non-parametric (Mann–Whitney) tests using the GraphPad Prism 8.2.1 software (GraphPad Software, Inc., La Jolla, CA, USA). A *p* value of <0.05 was considered statistically significant.

### 3.3. In Vivo Assays

Ethics Statement

Animal studies adhered strictly to the Italian Ministry of Health guidelines for the use and care of experimental animals (IACUC#733).

#### 3.3.1. Toxicity

Immunocompetent C57Bl/6NCr male mice (Charles River), 8–10-weeks-old (20–22 g), were utilized for toxicity evaluation and *P. aeruginosa* acute infection. All mice were maintained under specific pathogen-free conditions in sterile cages in a ventilated isolator. Toxicity of different doses of GY971a (pANDMA) was tested in C57BL/6NCr mice. GY971a (pANDMA) (4.5 mg/kg, 8 mg/kg and 18 mg/kg) or the vehicle (H_2_O/4%DMSO) were administered by aerosol by Penn Century 5 min after intratracheal (i.t.) inoculation of PBS (50 µL) to mimic the infection. 24 h after treatment, mice were monitored for behavior, weight loss, gross visual evaluation of health status and body temperature, then sacrificed, and different organs were excised for histopathological analysis.

#### 3.3.2. Mouse Models of Acute *P*. *aeruginosa* Infection

A model of acute infection with a *P. aeruginosa* PAO1 reference strain was used for efficacy studies. C57BL6/NCrl male mice (8–10-weeks-old) purchased by Charles River were anesthetized by an i.p. injection of 2.5% Avertin (0.015 mL/g body weight) and infected by intratracheal (i.t.) injection of planktonic 1 × 10^6^
*P. aeruginosa* bacterial cells [20,21]. To evaluate the efficacy of GY971a (pANDMA), mice were treated, 5 min after the infection, by local pulmonary administration using a Penn-Century MicroSprayer ^®^ Aerosolizer with different doses: 4.5 mg/kg, 9 mg/kg and 18 mg/kg or with the vehicle (H_2_O/DMSO 4%). CFU counts and cell counts in the lung and in the bronchoalveolar lavage fluid (BALF) were analyzed, as previously described [21,27,28], 6 h after acute infection. Cytokine/chemokine levels were measured in the supernatant of lung homogenates by Bioplex Assay (Bio-Rad Laboratories, Segrate, Italy).

Histopathological evaluation

At necropsy, bone marrow from sternum, heart, lungs with trachea, liver, spleen and kidneys were collected, fixed in buffered 4% formalin and embedded in paraffin. Hematoxylin and eosin-stained 3 μm paraffin sections were examined for histopathological analysis. Microscopic lesions were classified on a scale of 1 to 5 as minimal (1), mild (2), moderate (3), marked (4), or severe (5); minimal referred to the least extent discernible and severe the greatest extent possible. The assessment of the histopathology included an informal blinding step. The slides were reviewed unblinded, to determine the features, extension and severity of the pathology relevant to the disease process.

Statistical analysis

Statistical analyses were performed with GraphPad Prism (GraphPad Software, Inc., San Diego, CA, USA) using one-way ANOVA with Dunnet’s multiple comparison test. Outlier data, identified by Grubbs’ test, were excluded from the analysis.

## 4. Conclusions

The study of novel and innovative drugs for the treatment of CF lung disease is constantly evolving to ameliorate the clinical conditions of patients. In the conventional treatment of CF, the commonly utilized drugs are nonsteroidal anti-inflammatory drugs (NSAIDs) and steroid derivatives that possess, in addition to great benefits, many known side effects. The search for modern therapies to counteract the inflammation in CF patients is aimed at finding new potential anti-inflammatory drugs with alternative mechanisms of action that may replace or alternate the use of standard drugs.

In inflammatory processes, which involve patients with CF, NF-κB transcription factor plays a crucial role. Indeed, the expression of many genes encoding for cytokines, chemokines, adhesion molecules, and other proteins involved in inflammation is regulated by NF-κB. The NF-κB TF, following a cascade of intracellular biochemical events activated by extracellular stimuli, including cytokines as TNF-α, is left free to move into the nucleus, where it may activate specific pro-inflammatory genes. For this reason, it is extremely interesting to find new potential anti-inflammatory agents, which inhibit the action of NF-κB and the subsequent production of cytokines (particularly IL-8, a known biomarker of CF lung inflammation) [10].

During our research, we projected, synthesized and studied a new generation of TMA derivatives able to inhibit NF-κB/DNA interactions in vitro. Through EMSA experiments using a DNA target mimicking an IL-8 promoter, the TMA analogues able to inhibit the NF-κB/DNA interaction were selected to investigate their biological activity on CF IB3-1 cells. In this biological system, the NF-κB-dependent IL-8 expression was evaluated. The most active selected derivatives, GY964 (4-PhDMA) and GY971a (pANDMA), were finally tested in vivo, by using a murine model of acute infection with a *P. aeruginosa* PAO1 reference strain, used for efficacy studies. We demonstrated that GY971a was the best in vitro and in vivo TMA anti-inflammatory derivative. In fact, GY971a-treated mice showed a significant dose-dependent decrease in inflammatory cells recruited in the BALF, including neutrophils, indicating an evident reduction in inflammation; macrophages decreased significantly at higher doses too, without abnormal mice behavior detected, weight loss, body temperature or evident changes in the health status. At the same time, GY971a-treated mice showed a significant reduction in the typical chemokines, MCP-1 and MIP-1α, activated by bacterial infection. IL-1α and IL-1β, the pro-inflammatory cytokines that play a central role during infection were also significantly decreased in treated mice. Other cytokines/chemokines (IL-10, IL-12p40, IL-12p70, IL-17A, INF-γ, KC) were reduced after aerosol treatment, together with granulocyte–macrophage colony-stimulating factor (GM-CSF) and granulocyte colony-stimulating factor (G-CSF). These results indicate that a single dose of GY971a treatment by aerosol was effective in reducing acute airway murine inflammation induced by *P. aeruginosa* in a dose-dependent manner.

In addition, histopathological analyses were conducted to exclude toxicity and morphological changes in different tissues. Moreover, this molecule takes the advantage of being soluble in water and, therefore, of being able to be administered to animals through aerosol.

Previous studies in humans and murine models, including those from our group [29], have shown that therapeutic strategies that interfere with innate immune recruitment mechanisms have to be implemented with great caution since they harbor the risk of disabling innate host defense mechanisms and favoring risk of sepsis. Considering the efficacy of GY971a and its possible interference with innate immune recruitment mechanisms, we evaluated the possible risk of favoring bacterial infections. In the acute infection model, GY971a efficacy was associated with a modest increase in bacterial burden at the highest dose that was not statistically significant compared to the control, and these data indicate no risk of acute pulmonary exacerbation in a single-dose treatment.

Finally, the in vitro toxicity of the most interesting TMA derivatives was analyzed to evaluate possible side effects, analyzing the photoadduct formation with DNA under UVA irradiation [19], the IB3-1 cell growth and apoptosis, the anti-bacterial and the mutagenic effects [19]. As demonstrated, no phototoxic, mutagenic or cytotoxic effects were observed.

All the results obtained and described up to now seem to be very encouraging: we have identified one interesting molecule, GY971a (pANDMA), that is able to inhibit the expression of IL-8 in CF IB3-1 cells; it possesses anti-inflammatory activity mainly in vivo and does not appear to have toxic effects on all the utilized models. The challenge ahead is to take a step toward murine models of chronic infection, to further reflect the complexity of human diseases.

## Figures and Tables

**Figure 1 ijms-23-14483-f001:**
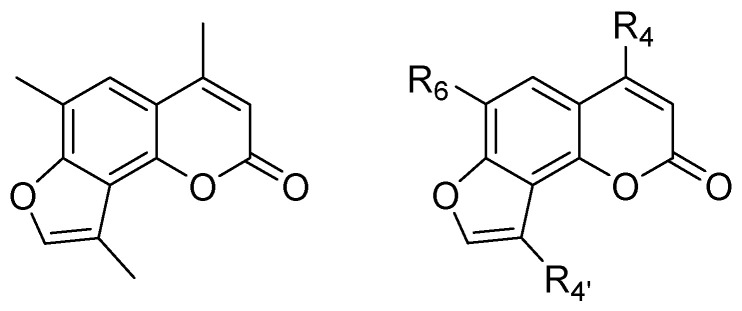
General structure of TMA and TMA analogues.

**Figure 2 ijms-23-14483-f002:**
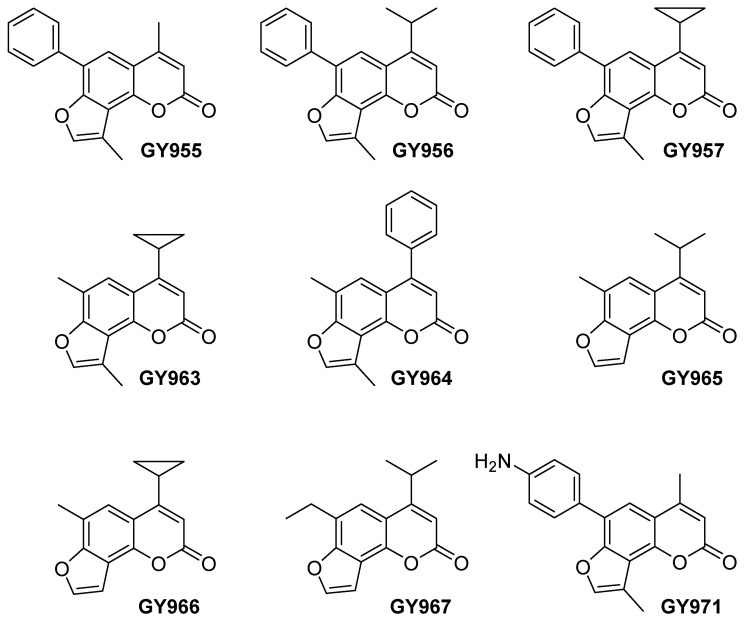
Chemical structures of the selected TMA analogues.

**Figure 3 ijms-23-14483-f003:**
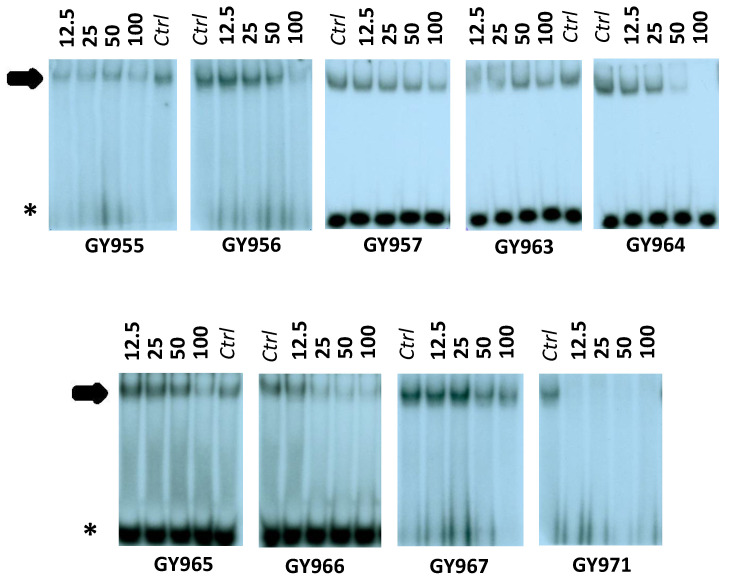
Effects of TMA analogues at 12.5–100 μM concentrations in EMSA experiments on the molecular interactions between NF-κB p50 and ^32^P-labeled target NF-κB double-stranded oligonucleotide. The arrow indicates NF-κB/DNA complexes; the asterisk indicates the free ^32^P-labeled target NF-κB probe.

**Figure 4 ijms-23-14483-f004:**
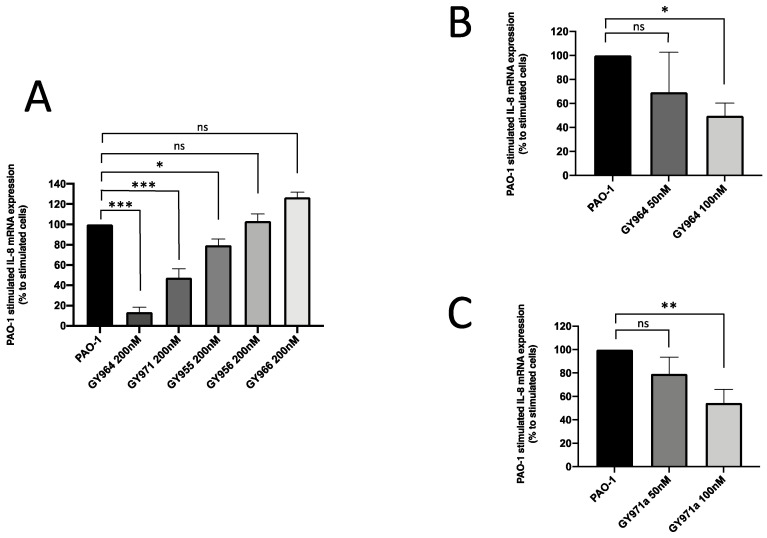
Effect of TMA analogues on IL-8 expression on PAO1 strain-infected IB3-1 cells. (**A**) Percent variations of IL-8 mRNA accumulation induced by 200 nM TMA analogues with respect to PAO1. (**B**,**C**) IL-8 mRNA quantity induced by TMA analogues GY964 (4-PhDMA) and GY971a (pANDMA) at different minor (50 and 100 nM) concentrations (mean values ± SD from three independent experiments). Statistics were performed by ANOVA, using Graph Pad Prism 9.0, where statistical significance at the levels of *p* < 0.05 (*) *p* < 0.01 (**), *p* < 0.001 (***) is reported, together with non-significance (ns).

**Figure 5 ijms-23-14483-f005:**
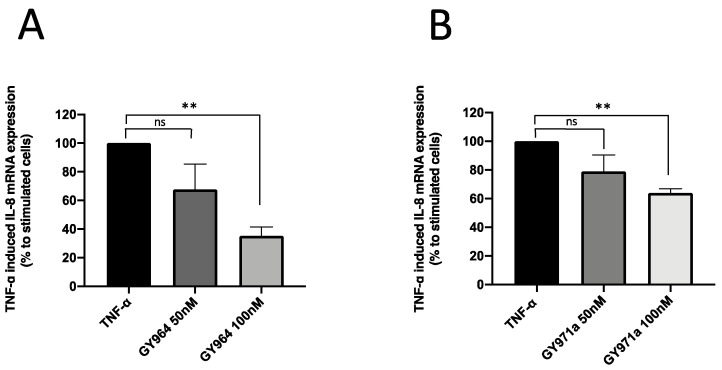
Effect of TMA analogues on IL-8 expression in TNF-α-stimulated IB3-1 cells. (**A**,**B**) Percent variations of IL-8 mRNA accumulation induced by GY964 (4-PhDMA) and GY971a (pANDMA) TMA analogues at different concentrations (50 and 100 nM) with respect to TNF-α-stimulated cells (mean values ± SD from three independent experiments). Statistics were performed by ANOVA, using Graph Pad Prism 9.0, where statistical significance at the levels of *p* < 0.01 (**) is reported, together with non-significance (ns).

**Figure 6 ijms-23-14483-f006:**
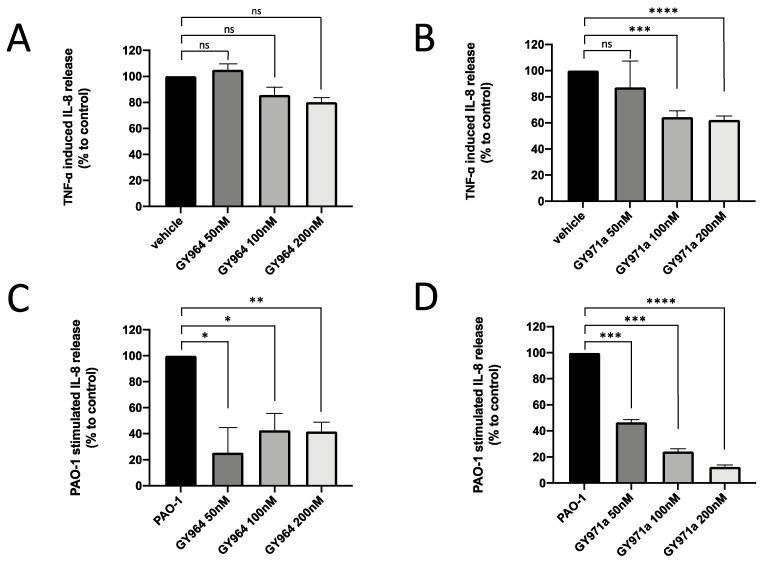
IL-8 release (% to control) in IB3-1 cells treated with different concentrations of GY964 (4-PhDMA) (**A**,**C**) and GY971a (pANDMA) (**B**,**D**) with respect to TNF-α-stimulated (**A**,**B**) or PAO1-infected (**C**,**D**) IB3-1 cells. Statistics were performed by ANOVA, using Graph Pad Prism 9.0, where statistical significance at the levels of *p* < 0.05 (*) *p* < 0.01 (**), *p* < 0.005 (***), *p* < 0.0001 (****) is reported, together with non-significance (ns).

**Figure 7 ijms-23-14483-f007:**
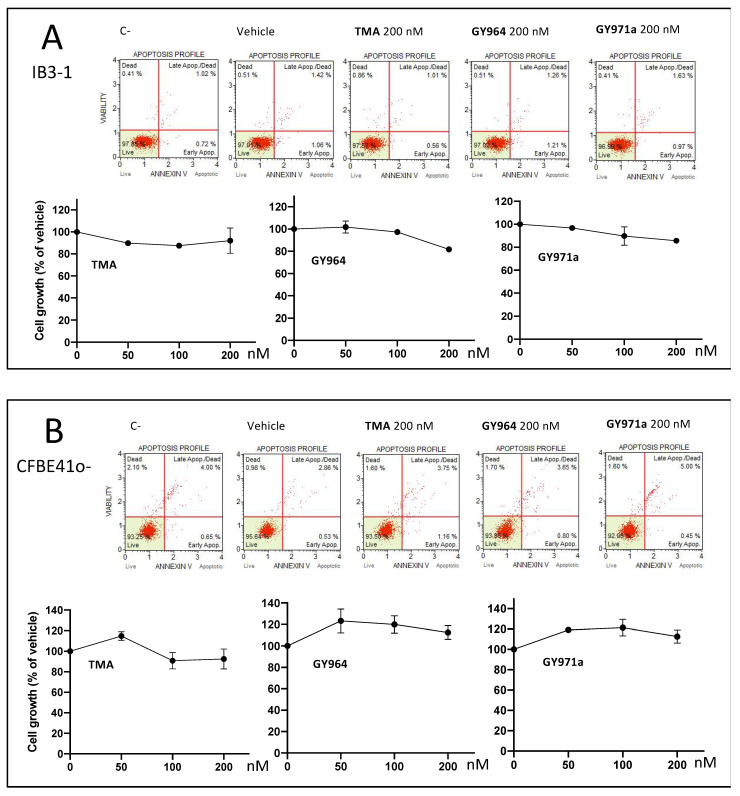
Representative data (apoptosis and cell growth) obtained from the treated IB3-1 (Panel **A**) and CFBE41o- (Panel **B**) cells: Muse Analyzer plots and cell count recorded 48 h after the treatment with 50, 100, 200 nM TMA, GY964 (4-PhDMA) and GY971a (pANDMA). Antibacterial activity of GY964 (4-PhDMA) and GY971a (pANDMA) (Panel **C**) against *Pseudomonas aeruginosa* (PAO1 strain) after 6 h of incubation (red dots), compared to the untreated strain (black dot) and the strain treated with Tobramycin 10 μM (TOBI) (green dot). C-: untreated cells; vehicle: DMSO.

**Figure 8 ijms-23-14483-f008:**
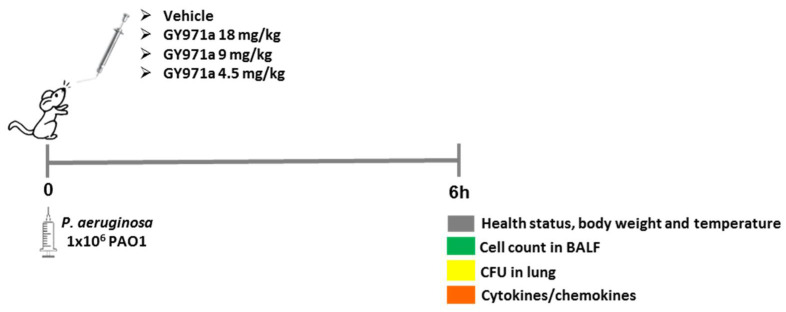
Scheme of the treatments and analysis in the murine model of *P. aeruginosa* acute infection.

**Figure 9 ijms-23-14483-f009:**
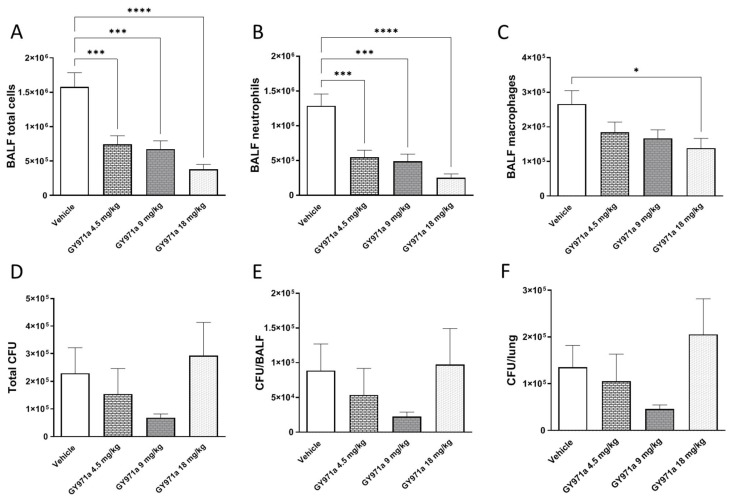
Efficacy of treatment with GY971a (pANDMA) in a murine model of acute lung infection. Mice were infected with 1 × 10^6^ colony-forming units (CFUs) of planktonic PAO1. 5 min after infection, 4.5 mg/kg, 9 mg/kg and 18 mg/kg GY971a (pANDMA) or the vehicle was administered as an aerosol by Penn Century. After 6 h, mice bronchoalveolar lavage fluid (BALF) was collected and the lungs were excised, homogenized and plated on tryptic soy agar to determine the bacterial burden. Total cell (**A**), neutrophil (**B**) and macrophage (**C**) counts were performed in bronchoalveolar lavage fluid (BALF). Bacterial burden was evaluated in lung (**D**), BALF (**E**) and in total lung (**F**). Data are presented as the mean ± sem pooled from two independent experiments n = 10–12. Statistics were performed by one-way ANOVA with Dunnet’s multiple comparisons test, using Graph Pad Prism 9.0. * *p* < 0.05, *** *p* < 0.001, **** *p* < 0.0001 compared to vehicle.

**Table 1 ijms-23-14483-t001:** MIC values for the TMA analogues in EMSA experiments.

Derivative	IUPAC Name (ID Name)	MIC ^1^
GY955	6-phenyl-4,4′-dimethyl-angelicin (6-PhDMA)	30 μM
GY956	4-isopropyl-6-phenyl-4′-methyl-angelicin (IPPhMA)	60 μM
GY957	4-cyclopropyl-6 phenyl-4′-methyl-angelicin (CPPhMA)	>100 μM
GY963	4-cyclopropyl-6,4′-dimethyl-angelicin (CPDMA)	100 μM
GY964	4-phenyl-6,4′-dimethyl-angelicin (4-PhDMA)	30 μM
GY965	4-isopropyl-6-methyl-angelicin (IPMA)	80 μM
GY966	4-cyclopropyl-6-methyl-angelicin (CPMA)	18 μM
GY967	4-isopropyl-6-ethyil-angelicin (IPEA)	>100 μM
GY971	6-*p*-aminophenyl-4,4′-dimethyl-angelicin (pANDMA)	<12.5 μM

^1^ Minimal Inhibitory Concentration. In the EMSA experiments, 12.5 μM was the lowest concentration used.

**Table 2 ijms-23-14483-t002:** Pro-apoptotic activity (%) of TMA and GY964 (4-PhDMA), dissolved in DMSO (Vehicle), and GY971a (pANDMA), dissolved in water, on IB3-1 cells.

Sample	Live	Early Apoptosis	Late Apoptosis	Dead	Total Apoptosis
Untreated cells (C-)	97.85	0.72	1.02	0.41	1.74
Vehicle	97.01	1.06	1.42	0.51	2.48
TMA 50 nM	96.20	0.65	1.30	1.85	1.95
TMA 100 nM	95.95	0.50	1.50	2.05	2.00
TMA 200 nM	97.57	0.56	1.01	0.86	1.57
GY964 50 nM	98.05	0.75	0.95	0.25	1.70
GY964 100 nM	97.55	0.50	1.20	0.75	1.70
GY964 200 nM	97.02	1.21	1.26	0.51	2.48
GY971a 50 nM	96.80	0.75	1.55	0.90	2.30
GY971a 100 nM	97.70	0.65	1.25	0.40	1.90
GY971a 200 nM	96.99	0.97	1.63	0.41	2.60

**Table 3 ijms-23-14483-t003:** Pro-apoptotic activity (%) of TMA and GY964 (4-PhDMA), dissolved in DMSO (Vehicle), and GY971a (pANDMA), dissolved in water, on CFBE41o- cells.

Sample	Live	Early Apoptosis	Late Apoptosis	Dead	Total Apoptosis
Untreated cells (C-)	93.25	0.65	4.00	2.10	4.65
Vehicle	95.64	0.53	2.86	0.98	3.38
TMA 50 nM	92.69	0.59	5.07	1.65	5.67
TMA 100 nM	93.40	0.70	5.00	0.90	5.70
TMA 200 nM	93.50	1.16	3.75	1.60	4.91
GY964 50 nM	94.80	0.50	3.25	1.45	3.75
GY964 100 nM	91.82	3.45	4.09	0.63	7.54
GY964 200 nM	93.85	0.80	3.65	1.70	4.45
GY971a 50 nM	94.20	0.95	3.20	1.65	4.15
GY971a 100 nM	94.75	0.70	3.25	1.30	3.95
GY971a 200 nM	92.95	0.45	5.00	1.60	5.45

**Table 4 ijms-23-14483-t004:** Cytokine and chemokine concentrations in supernatant of lung homogenates following acute *P. aeruginosa* airway infection and GY971a (pANDMA) treatments. Data are expressed as mean values ± standard errors of the means (SEM) pooled from two independent experiments (N = 8–10). Statistical significance is determined by one-way ANOVA with Dunnett’s multiple comparisons test and significant differences between treatments and the vehicle are highlighted in gray, using Graph Pad Prism 9.0. * *p* < 0.05, ** *p* < 0.01 *** *p* < 0.001 compared to vehicle.

Cytokine/Chemokine	Concentration pg × 700 µg^−1^			
	Vehicle (H_2_O/DMSO 4%)	4.5 mg/Kg GY971a	9 mg/kg GY971a	18 mg/kg GY971a
IL-1α	88.45 ± 10.30	65.95 ± 7.91	54.31 ± 7.29 *	43.51 ± 3.03 **
IL-1β	9.21 ± 1.49	6.95 ± 1.05	4.91 ± 0.59 *	4.89 ± 0.75 *
IL-2	6.27 ± 0.28	5.58 ± 0.30	5.76 ± 0.24	5.28 ± 0.20 *
IL-3	5.11 ± 0.40	4.58 ± 0.24	4.36 ± 0.41	3.67 ± 0.25 *
IL-4	1.29 ± 0.09	1.23 ± 0.12	0.93 ± 0.14	0.87 ± 0.11 *
IL-5	8.46 ± 0.68	8.13 ± 0.37	7.89 ± 0.63	7.26 ± 0.44
IL-6	110.0 ± 27.68	64.30 ± 10.86	49.64 ± 9.34 *	69.79 ± 8.27
IL-9	38.58 ± 1.01	35.43 ± 1.74	32.17 ± 2.04 *	29.64 ± 0.69 ***
IL-10	20.76 ± 0.95	17.55 ± 1.23	16.41 ± 1.56 *	14.79 ± 1.05 **
IL-12p40	71.29 ± 4.32	61.72 ± 5.31	52.88 ± 4.67 *	45.67 ± 2.62 ***
IL-12p70	167.6 ± 16.08	123.7 ± 12.91	110.1 ± 13.58 **	94.45 ± 6.61 **
IL-13	88.02 ± 3.81	82.53 ± 2.63	80.58 ± 4.30	78.14 ± 3.36
IL-17A	5.49 ± 0.63	4.27 ± 0.26	3.98 ± 0.30 *	3.84 ± 0.17 *
Eotaxin	1627 ± 160.8	1638 ± 124.4	1618 ± 87.33	1414 ± 65.28
G-CSF	444.1 ± 86.41	259.1 ± 36.89	297.8 ± 58.72	181.7 ± 19.88 *
GM-CSF	48.66 ± 5.94	45.70 ± 1.52	33.60 ± 4.60 *	42.55 ± 3.18
IFN-γ	30.10 ± 1.43	27.02 ± 1.51	23.76 ± 1.65 **	22.51 ± 0.96 **
KC	2413 ± 456.1	2085 ± 263.6	1486 ± 150.8	2048 ± 248.3
MCP-1	1813 ± 322.6	1171 ± 136.4	979.0 ± 144.3 *	692.2 ± 78.41 **
MIP-1α	186.9 ± 36.52	131.7 ± 23.94	107.0 ± 12.10	92.11 ± 10.33 *
MIP-1β	291.2 ± 30.26	254.0 ± 28.16	219.8 ± 20.41	212.2 ± 15.66
RANTES	110.3 ± 10.35	123.0 ± 9.93	109.9 ± 9.72	101.9 ± 9.40
TNF-α	104.6 ± 9.05	93.45 ± 6.93	95.61 ± 11.04	84.06 ± 5.22

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
