# Peer review of "New TMA (4,6,4′-Trimethyl angelicin) Analogues as Anti-Inflammatory Agents in the Treatment of Cystic Fibrosis Lung Disease"

_ijms, 2022, doi:10.3390/ijms232214483_

Round 1
Reviewer 1 Report
The manuscript by Chiara Tupini and co-workers describes the development of novel compounds derived from the trimethyl angelicin (TMA) structure and their effects on proinflammatory markers.
The authors utilise a panel of different methods such as electrophoretic mobility shift assay (EMSA), gene expression analysis and a murine acute airway infection model. The experiments demonstrate that the most potent substance, 6-p-aminophenyl-4,4’-dimethyl-angelicin (pANDMA), effectively inhibits the NF-κB/DNA complex and suppresses inflammation in the airway infection model.
The results are presented in a straightforward manner and the manuscript deserves publication. However, a few comments have to be made:
1. Table 3 shows the effect of 3 different concentrations of GY971a (pANDMA) on the concentration of various cytokines. While the authors have stressed the importance of IL-8 in the inflammatory pathway, this is the only cytokine that is not reported in table 3. Why did the authors chose to omit measurement of IL-8 in this context?
2. I assume that the inflammatory mouse airway model can also be utilised to test for more clinical outcomes such as survival etc.. Why did the authors not study the animal's outcome for a longer period. It would be worthwhile to report, whether an anti-inflammatory agent such as pANDMA has a positive or negative effect on the survival of the animals treated with this compound.
3. Minor point: Lines 272-275 are duplicate to lines 268-271. Please delete.
Author Response
Dear Professor,
Reviewer of our article regarding the development on new TMA analogues as possible anti-inflammatory agents in CF, we thank you for your work, analysis and suggestions. Point by point in red we here reported our responses to your appreciated comments:
“The manuscript by Chiara Tupini and co-workers describes the development of novel compounds derived from the trimethyl angelicin (TMA) structure and their effects on proinflammatory markers.
The authors utilise a panel of different methods such as electrophoretic mobility shift assay (EMSA), gene expression analysis and a murine acute airway infection model. The experiments demonstrate that the most potent substance, 6-p-aminophenyl-4,4’-dimethyl-angelicin (pANDMA), effectively inhibits the NF-κB/DNA complex and suppresses inflammation in the airway infection model.
The results are presented in a straightforward manner and the manuscript deserves publication. However, a few comments have to be made:
Table 3 shows the effect of 3 different concentrations of GY971a (pANDMA) on the concentration of various cytokines. While the authors have stressed the importance of IL-8 in the inflammatory pathway, this is the only cytokine that is not reported in table 3. Why did the authors chose to omit measurement of IL-8 in this context?”
The homologue of human IL-8 is completely absent from the genome of rodents, and in mice is compensated by KC. Thus, we have included KC in the panel of cytokines evaluated in vivo.
“I assume that the inflammatory mouse airway model can also be utilised to test for more clinical outcomes such as survival etc.. Why did the authors not study the animal's outcome for a longer period. It would be worthwhile to report, whether an anti-inflammatory agent such as pANDMA has a positive or negative effect on the survival of the animals treated with this compound”.
In our study, we test efficacy in an acute model of infection including microbiological and immunological readouts. General health was also monitored. The selected end-points are in accordance with European Respiratory Society task force recommendations (1) and respond to the predictive value and clinical relevance in drug testing. Measuring survival in mice of acute infection is not an appropriate end-point for ethical reasons. This approach needs frequent monitoring following an approved scoring system with predefined humane end-points. In our case, mice infected with P. aeruginosa should be sacrified before the termination of the study and this limit the analysis and statistical validation of the survival curve.
This study represents the first proof of concept for the efficacy of GY971a and we agree with the reviewer that the challenge ahead is to take a step toward murine models of chronic infection, to further reflect the complexity of human diseases. We have included this statement on page 15 (lines 423-425) at the end of the discussion.
- Bonniaud P, Fabre A, Frossard N, Guignabert C, Inman M, Kuebler WM, Maes T, Shi W, Stampfli M, Uhlig S, White E, Witzenrath M, Bellaye PS, Crestani B, Eickelberg O, Fehrenbach H, Guenther A, Jenkins G, Joos G, Magnan A, Maitre B, Maus UA, Reinhold P, Vernooy JHJ, Richeldi L, Kolb M. Optimising experimental research in respiratory diseases: an ERS statement. Eur Respir J 2018: 51(5).
“Minor point: Lines 272-275 are duplicate to lines 268-271. Please delete”.
Done.
Best regards,
Ilaria Lampronti
Reviewer 2 Report
Tupini et al reported the new TMA analogues as anti-inflammatory agents in the CF. Comparing with reported TMA, the major improvement they claimed is less toxicity. However, there are no directed comparison of their toxicities in vitro or in vivo. The demonstration of no “evident” toxicity is not solid since only one cell line was used and 24 h in vivo treatment time. A more comprehensive characterization of new TMA analogues ( in directed comparasion with TMA) should be included.
In Figure 6b, Why no inhibition of IL-8 release by GY964 treatment was observed in TNF-alpha induced IB3-cells?
“A slight increase in apoptosis percentage was observed only in relation to treatments 223 with GY964 200 nM, but this data probably was due to the contribution of the vehicle 224 (DMSO) used to solubilize this derivative insoluble in water, while GY971a was dissolved 225 in water.” This sentence is wrong. If it is the DMSP effect, the lower concentration of GY964 treatment should also exhibit around 18% apoptosis.
More cell lines should be used to evaluate the toxicity of selected TMA analogues. One cell line is not enough.
There is no reference at “Although drugs that target the CFTR have recently been approved and show great 82 promise, it is still not clear how CFTR-modulators affect theinfection and inflammation. “.
For in vivo toxicity study, the samples were collected after one day, which is too short for many toxicity to appear. Could you perform a long-time experiment to monitor its toxicities?
Could you run statistical analysis for Fig. 9D,E,F and indicate them in figure? Also, there should be 4.5 mg/kg instead of 4,5 mg/kg
In Figure 9 D,E,F, could you explain the CFU goes down when improving dose from 4.5 to 9 but goes up to even higher than vehicle when improving dose from 9 to 18? Could you explain why the drug is a promising therapy if the CFU increase?
Author Response
Dear Professor,
Reviewer of our article regarding the development on new TMA analogues as possible anti-inflammatory agents in CF, we thank you for your work, analysis and suggestions. Point by point in red we here reported our responses to your appreciated comments:
Tupini et al. reported the new TMA analogues as anti-inflammatory agents in the CF. Comparing with reported TMA, the major improvement they claimed is less toxicity. However, there are no directed comparison of their toxicities in vitro or in vivo. The demonstration of no “evident” toxicity is not solid since only one cell line was used and 24 h in vivo treatment time. A more comprehensive characterization of new TMA analogues (in directed comparasion with TMA) should be included.
The known major toxic effect of TMA (and linear psoralens) is related to its ability to bind, under UV irradiation, DNA strands, producing photoadducts (phototoxcicity) (Rousset et al. 1996), but we have very recently demonstrated, as reported by our research group (Vaccarin et al. IJMS 2022), that no photoadducts formation was observed in the analogues GY964 and GY971 in TLC analysis, in comparison with TMA (We added also a picture extracted from this article and reported in the “Supplementary material - Figure S1- to better explain this point). The publication (Ref. 19) is new and now we updated the reference (line 692). All the other possible cytotoxic effects on CF cell cultures were studied only in order to exclude any unexpected effects. To explain it, we added a sentence in the text (line 219) to underline this argument. In any case, to be more precise, as you rightly suggest, we tested the two analogues again by comparing them with the TMA (line 239) on two different models, IB3-1 and CFBE41o- CF cell lines (line 236) (the figure 7 was upgraded and new additional figures was reported in supplementary material -S2 and S3-; we added also a new Table 3, line 269, and a new paragraph in materials and methods, lines 553-560).
In Figure 6b, Why no inhibition of IL-8 release by GY964 treatment was observed in TNF-alpha induced IB3-cells?
We thank the Reviewer that underlined a very interesting question, that already puzzled us when writing the manuscript. First, we found that GY964 is able to reduce the IL-8 mRNA expression (Fig 5A) but not the IL-8 protein release (Fig 6B). This kind of discrepancy is not completely unexpected and that is why we try to check always the release of the protein as final evidence of a positive or negative effect of different molecules. Second, and more important, we agree that the discrepancy of the inhibitory effect produced by GY964 upon PAO1 or TNF-a challenges is puzzling (Fig 6C vs 6A new figure). Our tentative explanation takes into consideration that the pro-inflammatory intracellular signaling inducing IL-8 expression in bronchial epithelial cells by P. aeruginosa involves the MyD88 signaling downstream at least two Toll-like Receptors (Bezzerri et al. Bezzerri V, Borgatti M, Finotti A, Tamanini A, Gambari R, Cabrini G. Mapping the transcriptional machinery of the IL-8 gene in human bronchial epithelial cells. J Immunol. 2011;187(11):6069-81), which is quite different from the intracellular machinery activated by TNF-a with its receptor. Thus, it could be possible that GY964 inhibits a pathway that is bypassed by the TNF-a/TNFR intracellular signal transduction. We did not report these arguments in text since they seem to us too speculative. Moreover, from a translational point of view, we are proposing the GY971a mesylate salt as superior to GY964, also in view of its interesting effects not only in vitro but also in the murine model in vivo.
“A slight increase in apoptosis percentage was observed only in relation to treatments 223 with GY964 200 nM, but this data probably was due to the contribution of the vehicle 224 (DMSO) used to solubilize this derivative insoluble in water, while GY971a was dissolved 225 in water.” This sentence is wrong. If it is the DMSP effect, the lower concentration of GY964 treatment should also exhibit around 18% apoptosis.
Thank you for your observation: actually, in the table that describes the pro-apoptotic activity of GY964 we reported only the maximum volume utilized (5 mL of DMSO) of the working solution (40 mM) to treat 1 ml of cell cultures (24-well plates) and to obtain a 200 nM final concentration. To obtain 100 and 50 nM concentrations, 2.5 and 1.25 mL of 40 mM working solution were added, respectively. To eliminate this possible misunderstanding problem, we repeated the experiments using the same volume (2.5 mL of DMSO from 80, 40 and 20 mM working solutions) to obtain all the desired final concentrations. We have substituted the old table 2 with a new table, directly related with the new figure 7. And we added also a second table (Table 3) that in detail reports data from CFBE41o- treated cells, as required.
More cell lines should be used to evaluate the toxicity of selected TMA analogues. One cell line is not enough.
We started our in vitro screening on CF IB3-1 cells that we utilized also for the study of anti-inflammatory activity (IL8 expression and release), but to be sure of our presented results we improved, as suggested, the paper also with CFBE41o- cells. We added sentences in the text, new related figures (Figure 7 in the text and S2-S3 in the supplementary materials) and a new table (Table 3).
There is no reference at “Although drugs that target the CFTR have recently been approved and show great 82 promise, it is still not clear how CFTR-modulators affect the infection and inflammation”.
We added the following sentence and reference (15) in the text (lines 87-91) and in the bibliography (line 680):
“Although drugs that target the CFTR have recently been approved and show great promise, it is still not clear how CFTR-modulators affect the infection and inflammation and the research of novel anti-inflammatory agents have been obscured by the discovery of CFTR modulators, thereby inflammation represents a sort of orphan target in CF. Indeed, no anti-inflammatory agents specific for CF lung disease have been developed and those commercially available, steroidal and non-steroidal anti-inflammatory drugs, are characterized by known and important side effects”
Laselva, O.; Criscione, ML.; Allegretta, C.; Di Gioia, S.; Liso, A.; Conese, M. Like Growth Factor Binding Protein (IGFBP-6) as a Novel Regulator of Inflammatory Response in Cystic Fibrosis Airway Cells. Front Mol Biosci. 2022, 12, 9:905468
For in vivo toxicity study, the samples were collected after one day, which is too short for many toxicity to appear. Could you perform a long-time experiment to monitor its toxicities?
This study represents the first proof of concept for efficacy of GY971a. Here we tested toxicity and efficacy in short term acute infection model. We agree with the reviewer that the challenge ahead is to take a step toward chronic toxicity and murine models of chronic infection for efficacy studies. We have included this concept on page 19, lines 423-425, at the end of the discussion.
We can not proceed toward these goals in this paper as the ethical permission for animal experiments should be renewed with a different design and including different aims. This will be included in further studies.
Could you run statistical analysis for Fig. 9D,E,F and indicate them in figure? Also, there should be 4.5 mg/kg instead of 4,5 mg/kg
Statistical analysis was performed by comparing all groups. If it is not indicated, it is due to the lack of statistical significance.
In Figure 9 D,E,F, could you explain the CFU goes down when improving dose from 4.5 to 9 but goes up to even higher than vehicle when improving dose from 9 to 18? Could you explain why the drug is a promising therapy if the CFU increase?
Modestly increased bacterial burdens were recorded with a single administration of GY971a in acute infection at the highest dose but this was not statistically significant compared to the control. We have commented on this aspect in the manuscript on page 19, lines 403-412.
“Previous studies in humans and murine models, including those from our group (Döring et al. 2014), have shown that therapeutic strategies that interfere with innate immune recruitment mechanisms have to be implemented with great caution since they harbor the risk of disabling innate host defense mechanisms and favoring risk of sepsis. Considering the efficacy of GY971a and possible interference with innate immune recruitment mechanisms, we evaluated the possible risk of favoring bacterial infections. In the acute infection model, GY971a efficacy was associated with a modest increase of bacterial burden at the highest dose that was not statistically significant compared to the control and these data indicate no risk of acute pulmonary exacerbation in a single-dose treatment. To verify that reducing inflammatory cells in infected mice does not impair host defense or exacerbate infection, bacterial load was evaluated in the airways of mice, including BALF and lung. No significant increase in bacterial burden was detected in the CFUs between GY971a and vehicle”.
Best regards,
Ilaria Lampronti